# Molecular Characterization of the Nsp2 and ORF5s of PRRSV Strains in Sichuan China during 2012–2020

**DOI:** 10.3390/ani12233309

**Published:** 2022-11-26

**Authors:** Jun Zhao, Zhiwen Xu, Tong Xu, Yuancheng Zhou, Jiangling Li, Huidan Deng, Fengqing Li, Lei Xu, Xiangang Sun, Ling Zhu

**Affiliations:** 1College of Veterinary Medicine, Sichuan Agricultural University, Chengdu 611130, China; 2Chia Tai Animal Husbandry Investment (Beijing) Co., Ltd., Beijing 101301, China; 3Key Laboratory of Animal Diseases and Human Health of Sichuan Province, Chengdu 611130, China; 4Animal Breeding and Genetics Key Laboratory of Sichuan Province, Sichuan Animal Science Academy, Chengdu 610058, China; 5College of Animal Science, Xichang University, Xichang 615012, China

**Keywords:** porcine reproductive and respiratory syndrome virus, phylogenetic analysis, NADC34-like strain, Sichuan China

## Abstract

**Simple Summary:**

PRRSV is one of the important pathogens that seriously affects the pig industry. Sichuan is a major pig breeding province in China and lacks long-term continuous surveillance of PRRSV infection in pigs. In this study, the prevalence of PRRSV in Sichuan was investigated based on a total of 539 samples of tissues, blood and aborted fetuses from 13 pig breeding areas in Sichuan. The genetic diversity of the virus currently circulating in Sichuan during 2012–2020 was studied by analyzing the genetic variation of Nsp2 and ORF5 genes. The detection results indicated that PRRSV was ubiquitous in the swine production in Sichuan, with 52.32% positive rate. Phylogenetic analysis of ORF5s showed that the PRRSV strains sequenced in this study belonged to PRRSV-1 and PRRSV-2 (lineage 1.5, 1.8, 3.5, 5.1 and 8.7). The data available in this study suggested that the genomes of prevalent PRRSV strains were complex and diverse.

**Abstract:**

Porcine reproductive and respiratory syndrome virus (PRRSV) is an important pathogen that poses a serious threat to the global pig industry. Sichuan Province is one of the largest pig breeding provinces in China. There is a lack of reports on the continuous surveillance and systematic analysis of prevalent strains of PRRSV in Sichuan Province in recent years. To fill this gap, a total of 539 samples were collected from 13 breeding regions in Sichuan during 2012–2020. The detection result showed that the positive rate of PRRSV was 52.32% (282/539). The ORF5s and Nsp2 were obtained and further analyzed, with Chinese reference strains downloaded from the GenBank. Phylogenetic analysis showed that the PRRSV strains sequenced in this study belonged to PRRSV-1 and PRRSV-2 (lineage 1, 3, 5 and 8). In total, 168 PRRSV-2 strains were selected for ORF5 analyses, and these strains were classified into sub-lineage 8.7 (HP-PRRSV), sub-lineage 5.1 (classical PRRSV), sub-lineage 1.8 (NADC30-like), sub-lineage 1.5 (NADC34-like) and sub-lineage 3.5 (QYYZ-like), accounting for 60.71% (102/168), 11.31% (19/168), 18.45% (31/168), 2.97% (5/168) and 6.55% (11/168) of the total analyzed strains, respectively. The Nsp2 of identified PRRSV strains exhibited a nucleotide identity of 44.5–100%, and an amino acid identity of 46.82–100%. The ORF5 of the identified PRRSV strains exhibited a nucleotide identity of 81.3–100%, and an amino acid identity of 78.5–100%. A sequence analysis of ORF5 revealed that the mutation sites of GP5 were mainly concentrated in HVR1 and HVR2 and the virulence sites. In summary, the HP-PRRSV, NADC30-like PRRSV, Classic-PRRSV, QYYZ-like PRRSV, NADC34-like PRRSV and PRRSV-1 strains exist simultaneously in pigs in Sichuan. NADC30-like PRRSV was gradually becoming the most prevalent genotype currently in Sichuan province. This study suggested that PRRSV strains in Sichuan were undergoing genomic divergence.

## 1. Introduction

Porcine reproductive and respiratory syndrome virus (PRRSV) is the causative agent of porcine reproductive and respiratory syndrome (PRRS). It is an economically devastating pandemic disease of swine that mainly infects the respiratory system and reproductive system [1]. In 2006, there was a large-scale outbreak and epidemic of “unexplained fever” caused by highly pathogenic PRRSV (HP- PRRSV) with high morbidity and mortality in infected pigs, causing huge economic losses to the pig farming industry. [2]. PRRSV is a small enveloped, single-stranded, positive-sense RNA virus [3]. The viral genome is approximately 15 kb in length. PRRSV encodes at least 10 open reading frames (ORFs), comprising ORF1a, ORF1b, ORF2a, ORF2b and ORFs3–7 [4,5]. ORF1a and ORF1b encode viral replicase polyproteins that are then hydrolyzed into 16 mature non-structural proteins by the viral protease [6]. ORFs2–7 encode viral GP2–5, M, E and the nucleocapsid protein N [6]. The GP5 protein is one of the most variable proteins in PRRSV and is often used as one of the main bases for assessing the mutation of PRRSV viruses. Therefore, ORF5 is usually used for sequence analysis because of its high variability [7].

PRRSV can be divided into two genotypes, named PRRSV 2 (represented by VR-2332 strain) and PRRSV 1 (represented by Lelystad virus strain) [8]. PRRSV type 2 can be further divided into five genetic subtypes in China, including lineages 1, 3, 5, 8 and 9. Representative strains of lineage 1 contain NADC30, JL580, NADC34 and RFLP 1-4-4. Lineage 3 PRRSV is mainly prevalent in South China and has a low pathogenicity, and its representative strains include QYYZ and GM2. Lineage 5 mainly contains the classical PRRSV strain represented by VR-2332. Lineage 8 contains highly virulent strains represented by TJ, JXA1, TA-12 and classical strains represented by CH-1a. Lineage 9 was discovered in Xinjiang in 2011 [9,10]. Recently, PRRSV type 1 was reported in Fujian, Inner Mongolia, Beijing and Hong Kong [11,12,13]. With the emergence of HP-PRRSV, a series of biosafety control measures were adopted. Except for traditional vaccines based on strains such as CH-1R, new commercial vaccines based on strains, such as JXA1 and TJM, were developed and widely used. PRR was then effectively controlled. However, available data in recent years indicate that PRRSV remains one of the most common pathogens plaguing the Chinese pig industry, with a complex infection profile, high prevalence and highly genetically heterogeneous genetic and recombination characteristics, emphasizing the need for the continuous surveillance of PRRSV to develop effective control measures. [14,15,16,17,18].

Sichuan Province is the main pig producing area in China. Because of geographical factors, the incidence of pig epidemics in Sichuan Province always lags behind other regions in China. In Sichuan, NADC30-like PRRSV strains were first reported in 2016. In this study, we conducted a long-term investigation of PRRSV in Sichuan. The full-length ORF5 sequences and Nsp2 from 2012 to 2020 were analyzed. The focus of our analysis is the genetic relationship and variation characteristics between the detected strains in Sichuan and other PRRSV strains in China. This study can provide valuable insights into the PRRSV epidemic situation and the molecular characteristics of PRRSV strains in Sichuan, and can have important implications for the development of effective strategies for the prevention and control of PRRSV outbreaks.

## 2. Results

### 2.1. Prevalence of PRRSV in Clinical Samples in Sichuan during 2016–2020

A total of 539 clinical samples, including serum, hilar lymph node and lung, were collected from 13 breeding areas in Sichuan Province, from 2016 to 2020. The source information and detection results of the samples are summarized in Table 1. Among 539 clinical samples, 282 samples were identified in 52.32% (282/539) of the samples, including 156 JXA1-like PRRSV positive samples, 13 classic PRRSV positive samples, 13 NADC34-like PRRSV positive samples, 86 NADC30-like PRRSV positive samples, 10 QYYZ-like PRRSV and 4 PRRSV-1 positive samples. The positive detection rates of samples in 2016, 2017, 2018, 2019 and 2020 were 44.92%, 58.77%, 52.71%, 53.16% and 54.43%, respectively.

### 2.2. Sequence Alignment and Phylogenetic Analysis of ORF5 and Nsp2 HV

Phylogenetic analysis based on ORF5 demonstrated that the 169 PRRSV strains obtained in Sichuan Province during 2012–2020 were clustered into six different subtypes (Figure 1). Among 169 PRRSV strains, 102 PRRSV strains belonged to the sub-lineage 8.7 branch with highly pathogenic PRRSV represented by the JXA1 strain, 19 PRRSV strains belonged to the sub-lineage branch 5.1 with classic PRRSV represented by VR2332 strain, 31 PRRSV strains belonged to the sub-lineage branch 1.8 with NADC30-like PRRSV represented NADC30 strain, 11 PRRSV strains belonged to the sub-lineage 5.3 branch with QYYZ-like PRRSV strains represented by QYYZ strain, 5 PRRSV strains belonged to the sub-lineage 1.5 branch with NADC34-like PRRSV strains represented by NADC34 strain and 1 PRRSV strain belonged to type 1 (Figure 2).

### 2.3. Sequence Alignment and Phylogenetic Analysis of Nsp2 HV

Phylogenetic analysis based on Nsp2 demonstrated that the 106 PRRSV strains obtained in Sichuan Province during 2012–2020 belonged to six different subtypes (Figure 3). Among these 106 PRRSV strains, there was 1 type 1 PRRSV strain, 2 QYYZ-like strains, 11 VR2332-like strains, 4 NADC34-like strains, 22 NADC30-like strains and 66 JXA1-like strains. The results of the deduced amino acid identity analysis of the Nsp2 of PRRSV type 2 were shown in Table 2. The nucleotide identity of the PRRSV strains in Sichuan with the representative strains VR2332, QYYZ, NADC34, NADC30, JXA1 and RFLP 1-4-4 ranged from 50.4% to 99.8%, 50.7% to 99.5%, 47.7% to 98.2%, 39.5% to 98%, 44.5% to 100% and 46.1% to 92%., and the amino acid identity ranged from 27.1% to 99.7%, 30.1% to 99.7%, 26.4% to 96.6%, 36.6% to 95.9%, 39.7% to 100% and 26.7% to 89%, respectively. The results of the identity analysis showed significant variability in Nsp2 nucleotides and amino acids among different PRRSV strains in Sichuan.

The results of type 2 PRRSV ORF5 nucleotide and deduced amino acid identity analysis were summarized in Table 2. The nucleotide identity between PRRSV strain in Sichuan and the representative strains VR2332, QYYZ, NADC34, NADC30, JXA1 and RFLP 1-4-4 were 81.4% to 99.5%, 80.8–97%, 82.1–94.2%, 82.6–96.4%, 81.3–100% and 82.9–96%, respectively, and the amino acid identity was 78.5–99.5% 80.1–97.3%, 81–95%, 82–97%, 78.5–100% and 82–95.5%, respectively. The nucleotide and amino acid identity between the NADC34 strain and five NADC34-like strains in Sichuan were 94.2% and 96%, respectively, and the nucleotide and amino acid identity with RFLP 1-4-4 strain were 95% and 95.5%, respectively. The five NADC34-like strains in Sichuan were all derived from aborted fetuses from pig farms with large-scale abortions.

### 2.4. Amino Acid Mutation Analysis of GP5 and Nsp2 HV

There are 1 signal peptide, 2 hypervariable regions (HVR1 and HVR2), 3 transmembrane regions (TM1, TM2 and TM3) and 5 epitopes in type 2 PRRSV GP5 protein (Figure 4). The five epitopes include a decoy epitope, a neutral epitope (PNE) and three T-cell epitopes, as well as two virulence-related sites located at amino acids 13 and 151, respectively [19,20]. Representative strains of different lineages were selected for analysis of amino acid variation encoded by ORF5 with PRRSV strains in Sichuan from 2012 to 2020. As depicted in Figure 4, the variation of GP5 protein was mainly concentrated in HVR1 and HVR2. Compared to other NADC34-like strains, one amino acid site mutation (N32D) of NADC34-like PRRSV in Sichuan was located in the putative HVR1. Meanwhile, three substitution sites (A57N, N58K and K59S) were observed in NADC34-like PRRSV, and the virulence substitution sites (Q13R and K151R) were observed in JXA1 HP-PRRSV compared with most PRRSV strains of the other four lineages (Figure 4).

Figure 5 shows the amino acid sequences deduced from the hypervariable region (HV) of the Nsp2 of 6 reference strains (VR2332, JXA1, NADC30, NADC34, QYYZ and RFLP 1-4-4 strains) and 34 representative strains in Sichuan from 2012 to 2020. Compared with VR2332, NADC30-like PRRSV strains had 131 amino acid (aa) deletions in the Nsp2 hypervariable region, located at 323-433 aa, 483 aa and 503-521 aa, respectively, NADC34-like PRRSV strains had 100 aa deletions at 329-428 aa and HP-PRRSV strains had 30 aa deletions at 481 aa and 532-560 aa, respectively. Compared with VR2332, a classical PRRSV (CHSCSL-12017 strain), obtained in this study, has 21 aa deletions at 488-489 aa and 503-521 aa, which has the same deletion region as NADC30 strain, but it belongs to the VR2332 strain branch in the phylogenetic tree constructed based on Nsp2.

## 3. Discussion

PRRSV has been affecting the Chinese pig industry for more than 20 years since it was first reported in China in 1996 [21]. Vaccination is adopted in China to prevent and control the outbreak of PRRSV. Although vaccination has controlled PRRSV outbreaks to some extent, the uncontrolled use of highly pathogenic live attenuated vaccines has also contributed to the widespread spread of live attenuated vaccine virus in swine herds [22,23]. In recent years, the emergence of new mutants has made China’s PRRSV popularity more complicated. PRRSV strains in China were reported to show different patterns of recombination between members of lineages/sub-lineages with the emergence of new lineage 3 (QYYZ-like) viruses in 2010, sub-lineage 1.8 (NADC30-like) viruses in 2013 and sub-lineage 1.5 (NADC34-like) viruses [22,24,25]. Previous studies showed that PRRSV strains in Sichuan were undergoing significant genome changes, and different sub-lineage strains co-circulated in Sichuan [26,27]. In this study, a major shift in the predominant genotype was observed in Sichuan. Among the 282 positive samples, 170, 22, 86 and 4 samples were positive for HP-PRRSV, classic PRRSV, NADC30-like PRRSV and European PRRSV, respectively. The detection rate of the HP-PRRSV strain was the highest. The classical PRRSV and HP-PRRSV strains were the predominant circulating PRRSV strains in Sichuan from 2012 to 2017. The QYYZ-like and NADC30-like PRRSV strains were first reported in 2015 in Sichuan [28]. The NADC34-like strain (CHSCMY-22019) was first discovered in Sichuan in 2019, and four strains were detected in 2020. The above results indicated that multiple sub-lineages of PRRSV strains transmitted among pigs in Sichuan. The NADC30-like and NADC34-like PRRSV strains in lineage 1 were detected in the Sichuan pigs, which coincided with the evolution and epidemic time of PRRSV epidemic strains in China. Since 2012, HP-PRRSV detection rates are decreasing, while NADC30-like PRRSV detection rates are increasing.

Nsp2 and ORF5 have a high mutation rate and are also suitable as a phylogenetic and epidemiological marker. In this study, we analyzed the genetic evolution of Nsp2 hypervariable region and ORF5 of PRRSV derived from Sichuan during 2012–2020. HP-PRRSV strains were characterized by a discontinuous 30aa deletion (1aa and 29aa) in the Nsp2 coding region [29]. NADC30-like PRRSV showed three discontinuous deletions (111aa, 1aa, and 19aa) in Nsp2 compared to VR-2332 [22,30]. The NADC34-like PRRSV has continuous 100aa deletions in the Nsp2 coding region [31]. Sequence alignment showed that the five NADC34-like PRRSV strains detected in Sichuan had the same 100aa deletion of Nsp2 protein as the IA/2014/NADC34 strain identified and isolated from the United States in 2014.

The amino acids at positions 13 and 151 of the GP5 protein of highly virulent PRRSV strain generally were showed as R [32]. Previous studies have shown that amino acid 137 of the GP5 protein can also distinguish attenuated vaccine strains (A137) from wild strains (S137) [33]. In this study, we found that the 13th and 151th virulence sites of JXA1 HP-PRRSV are arginine (R), while most PRRSV strains of the other four lineages are glutamine (Q) and lysine (K), respectively, and a few are R. The amino acid at position 137 of the VR2332-like PRRSV strains, identified in this study, were from piglets with clinical signs. Based on the analysis of the GP5 protein in this study, mutations in the key amino acid sites of the currently popular wild strains may be one of the reasons for the poor effectiveness of the vaccine immunity. Interestingly, the phylogenetic tree analysis showed that the CHSCSL-12017 strain belonged to VR2332-like PRRSV strains, but it has continuous 19aa deletions at 502aa–521aa in Nsp2, which is similar to NADC30-like PRRSV. The ORF5 of the CHSCMY-12016 strain is in the same branch as the JXA1-like strains, while its Nsp2 is in the same branch as NADC30-like strains. The ORF5 of CHSCNC-22020 and CHSCMY-32020 strains belongs to the branch of QYYZ-like strains, while the Nsp2 belongs to the branch of NADC30-like strains. The ORF5 of the CHSCCD-42020 strain belongs to the branch of QYYZ-like strains, while the Nsp2 belongs to the branch of NADC34-like strains. These results suggested that the genome of PRRSV might be undergoing recombination.

In summary, we systematically analyzed the epidemic history and genetic evolution of PRRSV in Sichuan from 2012 to 2020 from the perspective of molecular epidemiology. The results of this study will provide valuable information and new horizons for PRRSV epidemic trends and control strategies.

## 4. Materials and Methods

### 4.1. Samples

A total of 539 samples from diseased pigs with several clinical manifestations (including respiratory symptoms, fever, abortion, etc.) were collected from swine suspected for PRRS on different pig farms in 13 breeding regions (Chengdu, Mianyang, Meishan, Luzhou, Yaan, Zigong, Leshan, Guangyuan and Suining). The sample type consisted of serum, hilar lymph nodes and lungs from pigs with suspected PRRS onset, which were collected during 2016–2020. Sample information, comprising sample type, geographical distribution and clinical manifestations is shown in Appendix A.

### 4.2. RT-PCR Detection and Amplification of Nsp2 and ORF5s

Approximately 5.0 g or 300 μL of each sample was homogenized separately in phosphate-buffered saline (PBS). According to the manufacturer’s instructions, viral RNA was extracted using the Viral RNA Extraction Kit (Takara, Dalian, China) and then the TIANScriptII RT Kit (TIANGEN BIOTECH (BEIJING) CO., LTD., Beijing, China) was used to acquire cDNA through reverse transcription. The primers of Nsp2 and ORF5 were designed based on the alignment of published PRRSV genome sequences obtained from the NCBI GenBank database (Appendix A). The cDNA was used for the PCR amplification of Nsp2 and ORF5. Then, the amplicons for each virus were submitted to Sangon Biotech Shanghai Co., Ltd., Shanghai, China for sequencing. The nucleotide sequences were determined for at least three independent cDNA clones.

### 4.3. Genetic Evolution Analysis of Nsp2 and ORF5s

All available GP5 and Nsp2 sequences of PRRSV in Sichuan China during 2012–2020 and other regions of China during 2012–2020 were downloaded from the GenBank database and then analyzed with the GP5 and Nsp2 sequences in this study. The reference sequences detail are shown in Appendix A. To explore the genetic characteristics of the dominant PRRSV strains in Sichuan from 2012 to 2020, a phylogenetic tree was constructed, based on the nucleotide sequences of Nsp2 HV and ORF5, using the neighbor-joining method in MEGA6.0 software package (http://www.megasoftware.net) with 1000 Bootstrap replicates of the alignment.

To determine the genetic relationship, the Nsp2 HV and GP5 sequences PRRSV in Sichuan from 2012 to 2020 were aligned with VR-2332, LV, JXA1, NADC30 and other reference sequences. Multiple sequence comparisons at the amino acid levels were performed by MEGA 6.0.

## Figures and Tables

**Figure 1 animals-12-03309-f001:**
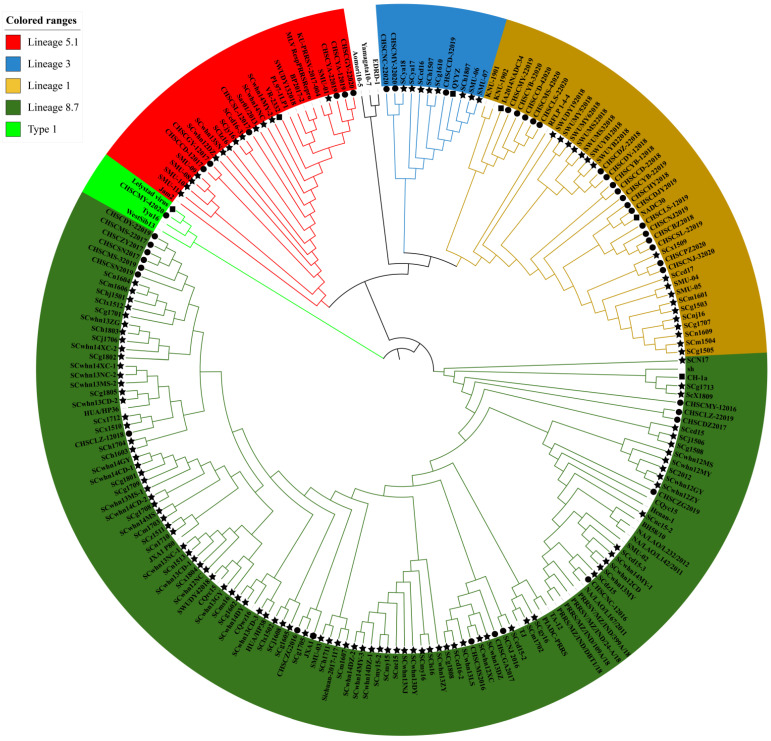
Phylogenetic analysis based on ORF5. The black dot represents PRRSV strains identified in this study, the black star diagram represents PRRSV strains from Sichuan in GeneBank from 2012 to 2020 and the black square represents PRRSV reference strains. Unmarked are current prevalent strains from other regions of China or other countries.

**Figure 2 animals-12-03309-f002:**
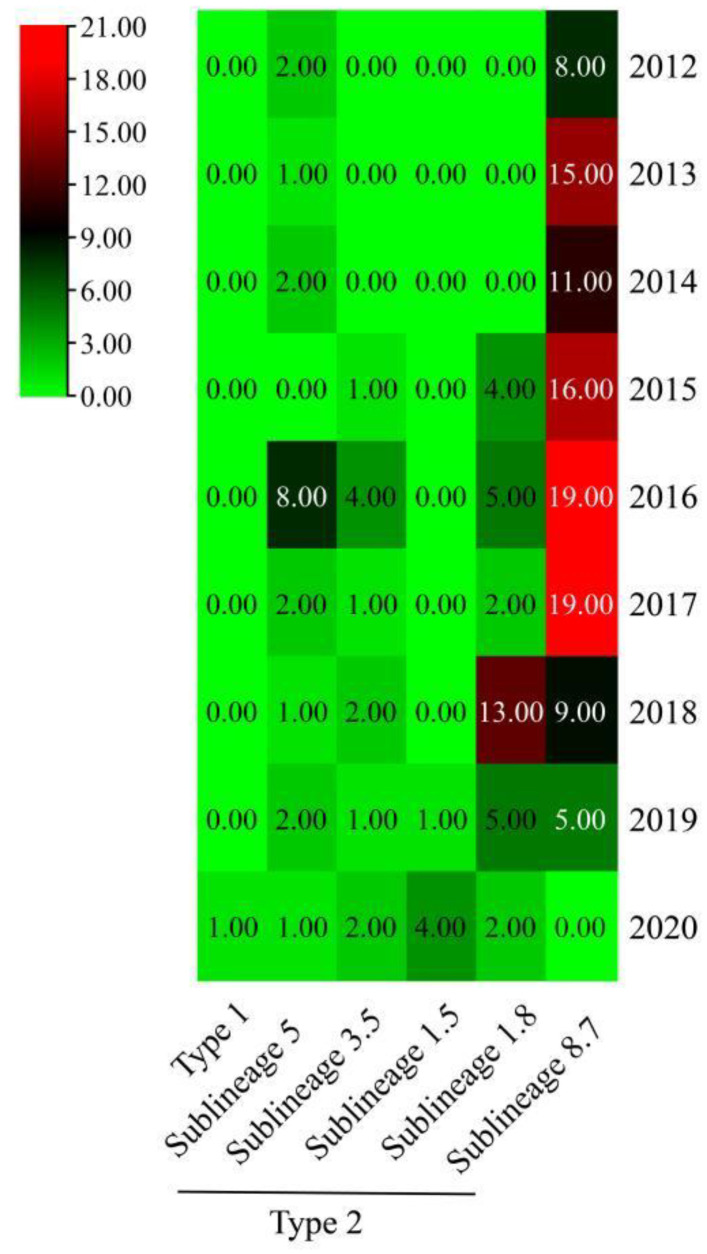
Heat map of the number of PRRSV strains in Sichuan from 2012 to 2020.

**Figure 3 animals-12-03309-f003:**
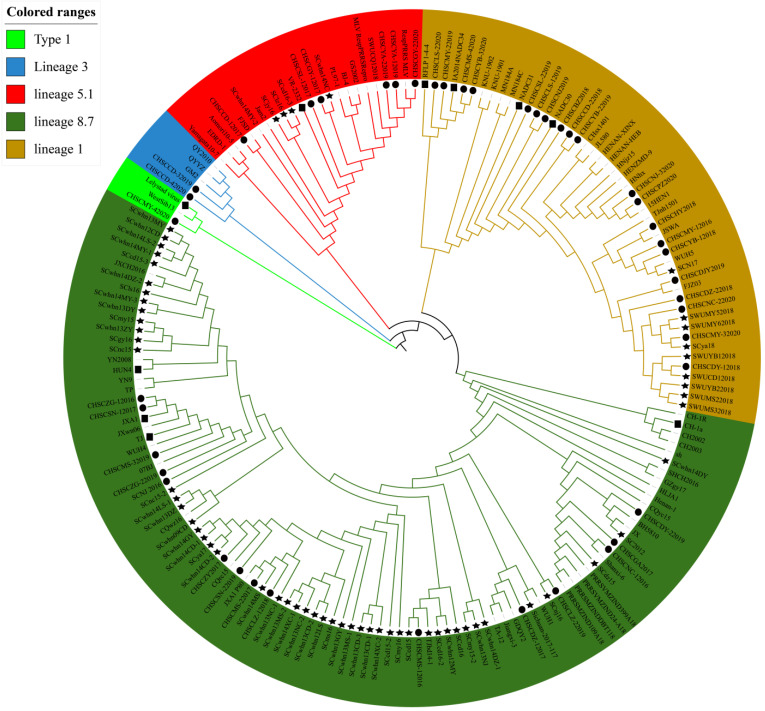
Phylogenetic analysis based on Nsp2. The black dot represents PRRSV strains identified in our study, the black star diagram represents PRRSV strains from Sichuan in Gene bank from 2012 to 2020 and the black square represents PRRSV reference strains. Unmarked are the current prevalent strains from other regions of China or other countries.

**Figure 4 animals-12-03309-f004:**
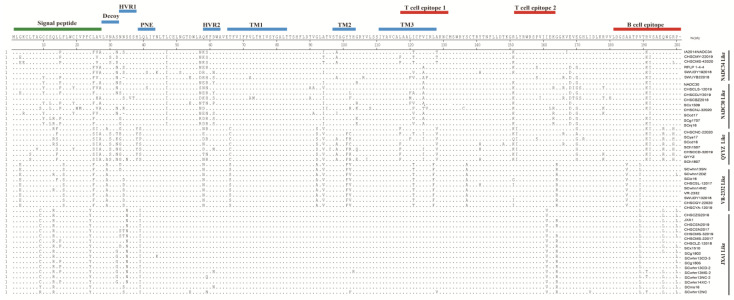
Amino acid sequence alignment of GP5 protein.

**Figure 5 animals-12-03309-f005:**
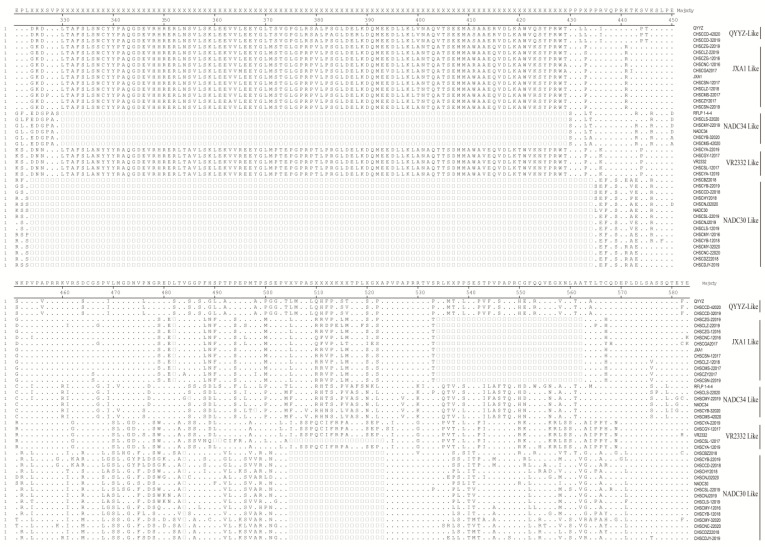
Amino acid sequence alignment of Nsp2 protein.

**Table 1 animals-12-03309-t001:** Statistics data of PRRSV positive samples.

Region	2016	2017	2018	2019	2020	Total	Positive Rate
Zigong	4/9	-	-	1/3	-	5/12	41.66%
Ziyang	2/10	4/11	-	-	3/4	9/25	36%
Yibin	3/20	12/20	4/6	3/7	4/4	26/57	45.61%
Yaan	-	-	13/23	6/7	-	19/30	63.33%
Suining	-	6/10	12/18	4/10	-	22/38	57.89%
Neijiang	13/28	-	7/10	3/8	3/14	26/60	43.33%
Nanchong	4/7	-	-	-	6/11	10/18	55.55%
Mianyang	10/13	15/24	4/17	2/7	7/7	38/68	55.88%
Meishan	6/14	6/7	2/7	3/3	2/4	19/35	54.28%
Luzhou	-	-	7/9	6/13	-	13/21	61.90%
Leshan	3/4	-	-	4/4	6/14	13/22	59.09%
Deyang	-	5/8	3/8	1/2	-	9/18	50.0%
Chengdu	17/33	4/11	6/15	9/15	7/12	43/86	50%
Guangan	-	10/13	-	-	-	10/13	76.92%
Dazhou	-	-	3/8	-	-	3/8	37.5%
Bazhong	-	-	7/8	-	-	7/8	87.5%
Guangyuan	-	2/2	-	-	5/9	7/11	63.67%
Total	62/138	67/114	68/129	42/79	43/79	282/539	52.32%
Positive rate	44.92%	58.77%	52.71%	53.16%	54.43%	52.32%	/

**Table 2 animals-12-03309-t002:** Nucleotide and deduced amino acid identity analysis based on ORF5 and Nsp2.

Reference Strains	ORF5	GP5 Protein	Nsp2 HVR	Nsp2 Protein HVR
VR2332	81.4–99.5%	78.5–99.5%	50.4–99.8%	27.1–99.7%
QYYZ	80.8–97%	80.1–97.3%	50.7–99.5%	30.1–99.7%
NADC34	82.1–94.2%	81–95%	47.7–98.2%	26.4–96.6%
NADC30	82.6–96.4%	82–97%	39.5–98%	36.6–95.9%
JXA1	81.3–100%	78.5–100%	44.5–100%	39.7–100%
RFLP 1-4-4	82.9–96%	82–95.5%	46.1–92%	26.7–89%

## Data Availability

The sequence data have been deposited in the NCBI GenBank under accession number MZ747395 to MZ747436 and MW915488 to MW915529.

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
