# Peer review of "Molecular Characterization of the Nsp2 and ORF5s of PRRSV Strains in Sichuan China during 2012–2020"

_animals, 2022, doi:10.3390/ani12233309_

Round 1

Reviewer 1 Report

In the manuscript of Zhao et al. the authors describe molecular evolution of PRRSV in the Sichuan province in China. A total of 539 samples were collected from 13 breeding regions in Sichuan during 2016-2020. The ORF5 and Nsp2 genes were sequenced analysed. PRRSV-2 strains belonged to the lineages 1, 3, 5 and 8, and one PRRSV-1 strain was found.

Comments.

Extensive edition of English is needed. Some of examples of the sentences that need correction can be found in lines 20-21, 23, 35-36, 44-45, 57, 77, 84, 89, 90, 108 and others.

Figure 1 and 3. It is difficult to differentiate black dots and black hexagons. Consider using different shapes or colours. There are some strains which not marked by any symbol. Why these strains were included into the analysis? Consider to exclude them.   

Figure 2 is not properly described in the text. What does this figure show?

Change term “homology” to “identity” all over the text. Homology indicates an ancient common origin and temporal evolution and refers to structural characteristics, whereas sequence identity is the amount of characters which match exactly between two different sequences.

Author Response

Comment 1: Extensive edition of English is needed. Some of examples of the sentences that need correction can be found in lines 20-21, 23, 35-36, 44-45, 57, 77, 84, 89, 90, 108 and others. Response: We appreciate the reviewer’s constructive comments. Extensive English have been edited, including lines 20-21, 23, 35-36, 44-45, 57, 77, 84, 89, 90, 108 and others. Comment 2: Figure 1 and 3. It is difficult to differentiate black dots and black hexagons. Consider using different shapes or colours. There are some strains which not marked by any symbol. Why these strains were included into the analysis? Consider to exclude them. Response: Thank you. Different shapes are used in Figures 1 and 3 for the reader to better distinguish. Unmarked are current prevalent strains from other regions of China or other countries, and the corresponding description has been added to the figure description and was marked in red. Comment 3: Figure 2 is not properly described in the text. What does this figure show? Response: Thanks for your suggestion. Regarding Figure 2, the number of virulent strains of each subtype has been described in detail in the article Comment 4: Change term “C” to “identity” all over the text. Homology indicates an ancient common origin and temporal evolution and refers to structural characteristics, whereas sequence identity is the amount of characters which match exactly between two different sequences. Response: Thank you. We have changed term “C” to “identity”.

Reviewer 2 Report

Zhao et al., evaluated the prevalence and molecular characteristics of porcine reproductive and respiratory syndrome virus (PRRSV) in the Sichuan Province of China. The authors evaluated a total of 539 samples collected from 13 breeding regions of Sichuan from 2016 to 2020. The study evaluated the number of PCR-positive samples and sequenced the ORF5 and Nsp2 genes to study the genetic evolution of PRRSV. The manuscript concluded that PRRSV in the Sichuan Province is as diverse as in other regions. 

The manuscript needs a lot of work in the English style. The abstract is very hard to understand and unclear and requires rephrasing.  The results section is also hard to follow, in part for the kind of results.  I can read a new clearly version. 

Figures require improvement, the quality is low. The number of samples analyzed in this study is low, considering that Sichiuan is the main pork-produced region in China.

The authors have to check the correct use of the term "homology" because, in my opinion, the correct one is "identity," at least in most cases. 

Line 43. endemic

Line 57: genotypes

Do the authors check all the ORF5 and Nsp2 sequences from each sample correspond to the same lineage? or there are samples with ORF5 corresponding lineage A and Nsp2 lineage B. It will be interesting if the authors can perform a recombination analysis. 

Author Response

Comment 1: The manuscript needs a lot of work in the English style. The abstract is very hard to understand and unclear and requires rephrasing.  The results section is also hard to follow, in part for the kind of results.  I can read a new clearly version.

Response: Thank you. We did a lot of work on the English style of the manuscript. The abstract and results sections were modified to make them easier to understand

Comment 2: Figures require improvement, the quality is low. The number of samples analyzed in this study is low, considering that Sichiuan is the main pork-produced region in China.

Response: Thank you. Figures have been improved. In fact, when we designed the study, the objectives were just to investigate the genetic diversity and prevalence of PRRSV in Sichuan China. So, when collecting clinical sample, we selected some samples suspected of PRRSV infection, which is the reason why the number of samples analyzed in this study is low.

Comment 3: The authors have to check the correct use of the term "homology" because, in my opinion, the correct one is "identity," at least in most cases.

Line 43. endemic

Line 57: genotypes

Response: Thanks for your suggestion. The homology has been changed to “identity”, and “endemic” in Line 43 and “genotypes” have been revised.

Comment 4: Do the authors check all the ORF5 and Nsp2 sequences from each sample correspond to the same lineage? or there are samples with ORF5 corresponding lineage A and Nsp2 lineage B. It will be interesting if the authors can perform a recombination analysis

Response: Thank you. The NSP2 and OFR5 sequences of each strain corresponded to each other, as shown in Table S3 and Table S4. Regarding recombination analysis, NSP2 and GP5 as single fragments in the genome are not suitable for recombination analysis. In the future, we will amplify the whole gene sequence and perform recombination analysis according to your suggestion.

Reviewer 3 Report

Zhao et al. investigated the epidemic history and genetic evolution of PRRSV in Sichuan from 2012 to 2020 from the perspective of molecular epidemiology. Some interesting results were obtained. But this study showed many limits and fundamental errors, with a limited discussion and conclusions. I added some comments and suggestions to Authors to improve the description before it can be considered for publication. My comments and suggestions were shown in more details below.

 Minor comments:

1. I suggested that the title of manuscript was changed with “Molecular Characterization of the Nsp2 and ORF5 Genes of PRRSV Strains in Sichuan China During 2012–2020” for a better description of the study. 2. Line 20-21: The sentence “The ORF5 gene…” is ungrammatical. Please check the statement. 3. Line 89-95: “test results” instead of “test result”. 4. Line 117: “ob tained” should be “obtained”.

5. Figure 1 and Figure 3: The labels are difficult to read, especially in the red part - the two figures should be improved.

6. It is not clear how viral sequences were selected from the GenBank database. 7. Figure 4:  I suggested using dot to replace the same amino acid to improve its readability.

8. The discussion is limited. The analysis should be performed on impact of other aspect, such as the amino acid changes on the potential lack of protection of the vaccine.

Author Response

Comment 1. I suggested that the title of manuscript was changed with “Molecular Characterization of the Nsp2 and ORF5 Genes of PRRSV Strains in Sichuan China During 2012–2020” for a better description of the study.

Response: Thanks for your suggestion. The title of manuscript has been changed with “Molecular Characterization of the Nsp2 and ORF5 Genes of PRRSV Strains in Sichuan China During 2012–2020”

Comment 2. Line 20-21: The sentence “The ORF5 gene…” is ungrammatical. Please check the statement.

Response: Thanks for your suggestion. The sentence has been corrected and marked with red color.

Comment 3. Line 89-95: “test results” instead of “test result”.

Response: Thanks for your suggestion. The sentence has been revised and marked with red color.

Comment 4. Line 117: “ob tained” should be “obtained”.

Response: Thanks for your suggestion. It has been revised.

Comment 5. Figure 1 and Figure 3: The labels are difficult to read, especially in the red part - the two figures should be improved.

Response: Thanks for your suggestion. The labels have has been modified to make it easier to read.

Comment 6. It is not clear how viral sequences were selected from the GenBank database.

Response: Sorry for the lack of clarity in the description of the viral genome sequence. More detailed information is added in the Materials and Methods section and marked in red.

Comment 7. Figure 4:  I suggested using dot to replace the same amino acid to improve its readability.

Response: Thanks for your suggestion. Figure 4 has been modified according to your suggestion.

Comment 8. The discussion is limited. The analysis should be performed on impact of other aspect, such as the amino acid changes on the potential lack of protection of the vaccine.

Response: Thanks for your suggestion. In the discussion section we add some other aspects of the analysis including “the amino acid changes on the potential lack of protection of the vaccine”, which was marked in red color.

Round 2

Reviewer 2 Report

The authors revised the manuscript and addressed the remarks. No further comments